# Rhythmic Dynamics of Stress Granules in *Wild-Type* and *Bmal1^−/−^* Fibroblasts Lacking a Functional Canonical Circadian Clock

**DOI:** 10.3390/ijms26209943

**Published:** 2025-10-13

**Authors:** Melisa Malcolm, Julio M. Pusterla, Laura G. Penazzi, Alejandra Trenchi, Victoria A. Acosta-Rodríguez, Maximiliano N. Ríos, Marcos Villarreal, Mario E. Guido, Eduardo Garbarino-Pico

**Affiliations:** 1Departamento de Química Biológica Ranwel Caputto, Facultad de Ciencias Químicas, Universidad Nacional de Córdoba, Cordoba X5000HUA, Argentina; mmalcolm@unc.edu.ar (M.M.); j.pusterla@fz-juelich.de (J.M.P.); gabriela.penazzi@unc.edu.ar (L.G.P.); victoria.acosta@gmail.com (V.A.A.-R.); maxi.rios@unc.edu.ar (M.N.R.); 2Centro de Investigaciones en Química Biológica de Córdoba (CIQUIBIC), CONICET, Universidad Nacional de Córdoba, Cordoba X500HUA, Argentina; 3Jülich Centre for Neutron Science, Forschungszentrum Julich GmbH, 52428 Julich, Germany; 4Instituto Multidisciplinario de Biología Vegetal (IMBIV), CONICET, Universidad Nacional de Córdoba, Cordoba X5000HUA, Argentina; atrenchi@imbiv.unc.edu.ar; 5National Institute on Aging, National Institutes of Health, Baltimore, MD 21224, USA; 6Instituto de Investigaciones en Físico-Química de Córdoba (INFIQC-CONICET), Departamento de Química Teórica y Computacional, Facultad de Ciencias Químicas, Universidad Nacional de Córdoba, Cordoba X500HUA, Argentina; mvillarreal@unc.edu.ar

**Keywords:** stress granules, circadian rhythms, biological clock, transcription–translation feedback loop (TTFL), BMAL1, fibroblasts, stress response, biocondensates

## Abstract

Circadian rhythms are endogenous ~24 h oscillations that regulate diverse biochemical processes. Although stress responses can exhibit circadian modulation, evidence for rhythmic regulation of stress granules (SGs)—cytoplasmic RNA–protein condensates formed under stress—remains limited. We investigated sodium arsenite-induced SG dynamics in NIH/3T3 cultures. SG number, eIF3 signal intensity—an established SG marker—and area oscillated with a period of ~24 h. These rhythms persisted in *Bmal1^−/−^* mouse embryonic fibroblasts (MEFs), despite lacking a transcription–translation feedback loop (TTFL) that constitutes the canonical circadian clock, but with altered amplitude and phase, indicating partial dependence on the molecular clock. Several SG-associated RNA-binding proteins (TIA-1, BRF1, hnRNP Q, and LARK) exhibited time-dependent changes at the mRNA and/or protein level, suggesting potential mechanisms for rhythmic SG modulation. Unlike previous in vivo reports linking SG variation to eIF2α phosphorylation, no temporal changes in phosphorylated eIF2α were observed, highlighting differences between isolated cells and tissues. Our results show that SG rhythmicity can persist without BMAL1, supporting alternative oscillatory mechanisms that contribute to the temporal organization of stress responses. Given their role in cell survival and the association of SG dysfunction with disease, these rhythms provide insight into how cellular stress responses are temporally regulated.

## 1. Introduction

Numerous biochemical, physiological, and behavioral processes exhibit cyclic changes with a period of approximately 24 h. While some of these variations arise as direct responses to daily environmental changes associated with the Earth’s rotation (e.g., light or temperature), many are generated by endogenous mechanisms known as circadian clocks [1,2]. These clocks are present in virtually all living organisms and play a crucial role in the temporal regulation of physiology and behavior. Although circadian oscillators are internally driven, they can be synchronized by external cues—such as light—to ensure alignment between internal rhythms and the external environment. Circadian rhythms are thought to confer adaptive advantages by allowing organisms to anticipate daily environmental fluctuations [3]. Among these fluctuations are environmental stressors such as high-intensity and ultraviolet light, which can generate oxidative stress and damage cellular components including DNA, proteins, and lipids. Such pressures are believed to have driven the evolution of circadian systems, favoring the temporal optimization of stress responses and antioxidant defenses to enhance efficiency at specific times of day [3,4]. Indeed, redox couples such as NAD/NADH, NADP/NADPH, and GSH/GSSG, as well as the stress response, are under circadian control in many cell types and tissues. Conversely, the redox state can also influence circadian clock function, indicating a bidirectional relationship between circadian timing and metabolism, particularly redox balance [5,6,7].

During the cellular stress response, cytoplasmic RNA–protein biocondensates known as SGs are formed. SGs are considered membraneless organelles, as they lack a lipid bilayer but are enriched in specific molecules and exhibit distinct physicochemical properties. They arise through liquid–liquid phase separation, typically triggered when global protein synthesis is inhibited as part of the stress response, leading to the accumulation of untranslated mRNAs and translation initiation factors. SGs contain mRNAs, translation initiation factors, small (40S) ribosomal subunits, RBPs, and various regulatory and scaffolding molecules [8,9]. Although their precise functions are not yet fully understood [10], SGs are known to regulate mRNA translation, storage, and stability. Importantly, SG formation has been associated with increased cell survival under adverse conditions. Moreover, SGs can act as hubs for cell signaling by sequestering and modulating the activity of signaling proteins, thereby influencing pathways involved in cell survival, apoptosis, and other cellular processes. Defects in SG assembly or clearance have also been implicated in a variety of pathological states, including neurodegenerative diseases, cancer, and viral infections [8,9].

As noted earlier, the stress response has been shown to exhibit circadian modulation in various organisms and cell types. While a single study has reported daily fluctuations in SG abundance in mouse liver, this evidence remains limited and not fully conclusive [11]. In addition, processing bodies (PBs)—another class of cytoplasmic mRNA biocondensates related to SGs—have been shown to exhibit circadian rhythmicity in U2OS and Neuro2a cell cultures [12,13]. Various factors influencing biocondensate formation, including daily changes in RNA and protein content, rhythmic levels of specific molecules, and fluctuations in ions and other metabolites, exhibit circadian oscillations, discussed in [14]. Based on these findings, we hypothesized that SG dynamics may also be modulated by circadian clocks. To test this, we investigated the temporal regulation of SGs induced by sodium arsenite in synchronized NIH/3T3 fibroblast cultures. We found that SGs detected by anti-eIF3 immunostaining displayed rhythmic changes in number and size, with a period consistent with circadian regulation. Furthermore, mRNA levels of several SG components, including TIA-1—a key nucleating factor—also showed significant temporal variation. Notably, SG oscillations persisted in *Bmal1^−/−^* fibroblasts, which lack a functional core circadian clock, suggesting the involvement of noncanonical circadian mechanisms.

## 2. Results

### 2.1. Stress Granule Number Varies over Time in Fibroblasts Synchronized by Serum Shock

To determine whether SG formation follows a temporal pattern, we examined SG abundance in NIH/3T3 fibroblast cultures over time. Fibroblasts have been widely used in chronobiology studies since the seminal work of the laboratory of Ueli Schibler [15]. They contain functional circadian clocks that regulate gene expression [16] and, as described by our group, circadian rhythms in metabolism [17,18]. Cell cultures were synchronized with a 2 h serum shock to align the circadian phase of individual oscillators [16,17,18]. SGs were induced by sodium arsenite treatment (500 μM for 30 min) at different time points across a 28 h window and detected by immunocytochemistry (ICC) using an anti-eIF3 antibody, a well-established SG marker [19,20,21].

To avoid confounding effects of mitotic synchronization, cells were grown to confluence and maintained for an additional 24–48 h in 0.5% serum. Under these conditions, they enter a quiescent (G_0_) state, as previously confirmed by flow cytometry [18]. As expected, untreated control cells showed no SGs, with eIF3 displaying a diffuse cytoplasmic distribution consistent with its role in translation initiation complexes (Figure 1A). In contrast, arsenite-treated cells formed numerous SGs (Figure 1). SG quantification was performed using ImageJ/Fiji 1.54k (see Section 4), and Figure 1A shows representative images with the segmented SGs overlaid as marks on the original micrographs, along with the negative control without primary antibody.

The number of SGs per image/field varied significantly over time (Figure 1B), peaking at 524 ± 39 SGs/field at 28 h post-synchronization compared to 140 ± 20/field at 14 h. One-way ANOVA revealed a highly significant effect of time (*p* < 0.001, *F* = 37.7; *n* = 17–19 images per group). The average number of nuclei per field (10.36, based on Hoechst staining) allowed estimation of approximately 26 SGs per cell across all time points. This estimate should be considered relative, as variations in the intensity threshold used for image segmentation can change the absolute value. However, the same segmentation parameters were applied consistently to all images from each experiment, ensuring unbiased comparisons.

SG signal intensity, calculated as the average SG fluorescence corrected for background, also exhibited time-dependent variation, with peaks at 7 h and 28 h (72 ± 0.7 at 28 h vs. 66 ± 0.6 at 21 h; ANOVA *p* < 0.001, *F* = 23.4). Although the amplitude of change was modest (~9%), the effect was statistically significant, suggesting dynamic regulation of eIF3 accumulation within SGs. SG area also showed small but significant changes (ANOVA *p* < 0.001, *F* = 11.1), with an 18% difference between peak (28 h) and trough (21 h). Area values were expressed in pixels^2^, reflecting relative measurements dependent on image processing parameters (see Section 4), although absolute values were consistent with previously reported SG sizes.

These findings were reproducible in two independent experiments. Notably, SG number, intensity, and area exhibited similar temporal profiles, indicating coordinated temporal regulation of SG formation and properties in fibroblasts synchronized by serum shock.

### 2.2. Temporal Changes in Stress Granules Are Not Attributable to eIF3 Expression or eIF2α Phosphorylation

The temporal variations in SG abundance described above could potentially reflect differences in the expression levels of the SG marker eIF3 used for their visualization. To examine this possibility, we analyzed eIF3 protein levels in synchronized NIH/3T3 cultures treated with sodium arsenite exactly as described in Figure 1. Western blot analysis of two independent experiments revealed no detectable differences in eIF3 expression at any time point (Figure 2).

Because phosphorylation of eIF2α is one of the earliest events involved in SG induction in response to stress [22], we next evaluated whether p-eIF2α levels vary over time. We hypothesized that time-dependent regulation of eIF2α phosphorylation could underlie the oscillations in SG number and properties. To test this, we measured p-eIF2α by Western blot across the same time points in control and arsenite-treated synchronized cells. As shown in Figure 2, a representative blot from two independent experiments demonstrated no noticeable temporal differences in p-eIF2α levels.

These results indicate that the temporal regulation of SGs is not attributable to variations in eIF3 protein levels or eIF2α phosphorylation. Instead, the oscillations likely occur downstream of eIF2α phosphorylation or via alternative regulatory pathways.

### 2.3. Several RNA-Binding Proteins That Are Components of RNA Granules Exhibit Temporal Regulation

To identify candidate factors that could account for the temporal changes in SGs described above, we searched the literature and the CircaDB database of rhythmic gene expression [23] for cytoplasmic RNA granule components reported to be circadianly regulated. Reverse transcription quantitative PCR (RT-qPCR) (Figure 3) and Western blot (Figure 4) analyses were performed to assess temporal regulation of candidate RBPs in serum-synchronized, quiescent NIH/3T3 cultures (samples every 7 h for 56 h).

As a control for synchronization, we confirmed the expected oscillatory expression of the clock gene *Bmal1* by RT-qPCR (ANOVA *F* = 4.07, *p* = 0.01, *n* = 3/group; Appendix A) (Figure 3). *Brf1* mRNA showed robust oscillations with peaks at 28 h and 56 h post-synchronization (ANOVA *F* = 4.48, *p* = 0.0136, n = 3 per time point; Appendix A). *Lark1* mRNA peaked at 28 h (ANOVA *F* = 18.10, *p* < 0.0001; Appendix A), while *Lark2* displayed peaks at 28 h and 49 h (ANOVA *F* = 23.49, *p* < 0.0001; Appendix A). Both *hnRNP Q* isoforms peaked at 7 h (*hnRNP Q1*: ANOVA *F* = 89.63, *p* < 0.0001; Appendix A. *hnRNP Q2*: ANOVA *F* = 46.19, *p* < 0.0001; *n* = 3/group; Appendix A). *Tia-1* transcripts exhibited strong oscillations with maxima at ~21–28 h and ~49 h (ANOVA *F* = 12.28, *p* < 0.0001; *n* = 3/group; Appendix A).

TIA-1 was detected by Western blot as two immunoreactive bands (upper band B1 and lower band B2) (Figure 4); both bands followed a similar temporal pattern, with minima at 14 h and maxima at 28 h post-synchronization. Comparison of control versus sodium-arsenite-treated samples showed higher TIA-1 levels after arsenite at most time points, with the largest treatment-dependent increase observed for the B1 band at 14 h (Figure 4). LARK protein did not show temporal changes under basal conditions but was markedly increased by arsenite treatment at all time points examined (Figure 4).

In summary, several RBPs involved in RNA granule biology (BRF1, hnRNP Q, LARK, and TIA-1) display temporal changes in their expression in synchronized NIH3T3 fibroblasts. Among them, TIA-1 stands out because of its well-established role in SG nucleation and its temporal profile mirroring that of the SG parameters analyzed.

### 2.4. Stress Granule Temporal Changes Are Rhythmic

To determine whether SG temporal variations are rhythmic, we monitored them for 68 h in dexamethasone-synchronized NIH/3T3 cultures (Figure 5). Three parameters were quantified: number, mean signal intensity, and area (Figure 5). All varied significantly over time (Table 1; see also Appendix A for post hoc comparisons). MetaCycle analysis [24] (see Section 4) indicated rhythmicity for all three parameters (Table 1). The oscillation period was 24 h for SG number and 20 h for intensity and area. SG number showed the most robust rhythm, whereas intensity had a lower amplitude and area, upon visual inspection, did not display a clear rhythmic pattern. Nevertheless, MetaCycle confirmed their statistical significance. Time-series data were also examined using Singular Spectrum Analysis, a model-free statistical tool that decomposes signals into trend and oscillatory components [25]. This analysis revealed a dominant low-frequency trend and a secondary oscillatory mode with a period of ~24 h and a modulus close to 1, consistent with a stable circadian component despite modest variance contribution (Appendix A). Overall, these results suggest that SG dynamics in synchronized NIH/3T3 fibroblasts are under circadian regulation.

### 2.5. Stress Granule Oscillations Persist in Cells with Impaired Canonical Molecular Circadian Clock

To determine whether the molecular circadian clock drives the observed SG changes, we analyzed these biocondensates in fibroblasts lacking functional clock machinery. Mouse embryonic fibroblast (MEF) cell lines wild-type (*wt*) and *Bmal1^−/−^*, both expressing the PERIOD2::LUCIFERASE (PER2::LUC) reporter, were used [26,27]. *Bmal1* knockout mice exhibit arrhythmic behavior and disrupted clock gene transcription [26]. BMAL1 is an essential, non-redundant component of the canonical TTFL regulating mammalian circadian rhythms [1,2,26].

We quantified SG number, signal intensity, and area in quiescent *wt* and *Bmal1^−/−^* fibroblasts synchronized with dexamethasone over 56 h (Figure 6). Both cell lines showed significant temporal changes in all parameters (Table 1 and Appendix A). MetaCycle analysis revealed rhythmicity in SG number, intensity, and area for both *wt* and *Bmal1^−/−^* fibroblasts (Table 1). The estimated period for SG number was similar between *wt* and *Bmal1^−/−^* (34.5 h) and *Bmal1^−/−^* (35 h) cells, with a phase difference of approximately 5.5 h. It should be noted that period estimates are limited by the sampling frequency used in the experiments, which affects precision. The amplitude of the rhythm was about 15% greater in *wt* cells (Table 1). Intensity and area showed lower amplitude rhythms in both lines, consistent with previous observations in NIH/3T3 fibroblasts, but were still classified as rhythmic by MetaCycle. Period differences between *wt* and *Bmal1^−/−^* cells were more pronounced for intensity and especially area (Table 1).

To confirm disruption of the TTFL in *Bmal1^−/−^* cells, we measured PER2::LUC bioluminescence in dexamethasone-synchronized cultures (Figure 6C, Table 1 and Appendix A). As expected, *wt* fibroblasts displayed robust circadian bioluminescence rhythms, while *Bmal1^−/−^* cells showed no rhythmicity and substantially lower signal levels, consistent with the role of BMAL1 in activating *Period2* transcription through heterodimerization with CLOCK or NPAS2.

## 3. Discussion

This study demonstrates that SGs induced by sodium arsenite, a potent oxidative stressor, undergo rhythmic changes in NIH/3T3, *wt* and *Bmal1^−/−^* fibroblast cultures. SG number, eIF3 signal intensity, and area fluctuated over time, as did several SG-associated components. Oscillations persisted in *Bmal1^−/−^* cells, indicating that these rhythms can occur even without a functional molecular clock. Although differences in rhythmic parameters between immortalized NIH/3T3 cells and MEFs may reflect cell type-specific properties, the restricted sampling frequency implies that period estimates are only approximate and should therefore be interpreted with caution.

SG dynamics were observed under cell cycle arrest and stable culture conditions, thereby ruling out cell cycle progression or external cues as primary determinants. Although initially attributed to the molecular clock, these rhythms persisted in *Bmal1* knockout cells, *Bmal1* being an essential and non-redundant core clock gene, suggesting the involvement of alternative oscillatory mechanisms.

Wang et al. [11] reported daily changes in the proportion of SG-containing cells in mouse liver after sodium arsenite injection. As the animals were kept under a light/dark cycle, it remains unclear whether the observed 20 h pattern was driven by the endogenous circadian system. Nonetheless, their findings align with our observations of temporal variation in stress-induced SG formation. Wang et al. [11] proposed that SG variations were due to oscillations in eIF2α expression. In contrast, we detected no significant temporal changes in p-eIF2α in fibroblast cultures after arsenite treatment. This discrepancy could reflect regulation of liver eIF2α by systemic signals absent in isolated cells. Supporting this, Kornmann et al. [28] and Koronowski et al. [29] showed that many liver gene oscillations depend on systemic circadian cues. Other reports have also described rhythms in eIF2α phosphorylation in mouse suprachiasmatic nuclei [30] and in *Neurospora*, where such rhythms are associated with translational rhythms [31,32,33]. Castillo et al. further linked eIF2α phosphorylation to rhythmic translation via mRNA localization to *Neurospora* cytoplasmic mRNP granules. However, direct comparison between in vivo systems and our cell culture model remains challenging, and the differences observed may arise from the distinct nature and complexity of the experimental systems.

Wang et al. [11] also examined the short-term dynamics of SG formation (induced by 30 min arsenite exposure) and clearance over 150 min in SH-SY5Y neuroblastoma cells. They reported that *Bmal1* knockdown increased the proportion of SG-positive cells and enhanced SG dynamics, as assessed by FRAP using the GFP–G3BP1 marker, while concomitantly reducing apoptosis. Conversely, *Bmal1* overexpression produced the opposite effects. Although this connection remains speculative, the propensity of cells with reduced BMAL1 levels to form more SGs could be related to our observation of a greater amplitude in SG rhythmicity in *Bmal1^−/−^* cultures. However, in apparent contradiction to this, the average number of SGs across all time points was higher in wild-type cells. On the other hand, Wang et al. also reported that, when similar experiments were conducted in *Bmal1^−/−^* and *Nr1d1^−/−^* (*Rev-Erbα*) MEFs, no differences were observed in the proportion of SG-containing cells.

The temporal regulation of several RBPs identified here—BRF1, hnRNP Q, LARK, and TIA-1—provides potential mechanisms for rhythmic modulation of SGs. TIA-1 is a well-characterized SG nucleator, with a prion-like Q-rich domain and RNA-binding motifs that promote condensation and translational silencing [19,22]. The synchronized oscillation of *Tia-1* mRNA and both TIA-1 protein bands (B1/B2), paralleling SG number and property changes, suggests that time-of-day variation in TIA-1 abundance or modification state could influence nucleation propensity. The presence of two protein bands is consistent with the existence of TIA-1 isoforms generated by alternative splicing [34], both of which display similar rhythmic profiles in our assays. BRF1 (ZFP36L1), an ARE-binding RBP involved in mRNA decay and PB dynamics, has been proposed to mediate interactions between PBs and SGs [35,36,37]; rhythmic *Brf1* expression may thus regulate mRNA fate. hnRNP Q (SYNCRIP/NSAP1) is a rhythmically expressed RBP that regulates translation of several clock genes [38,39,40,41,42,43] and has been reported as a component of RNA granules [44,45,46]. LARK (RBM4) is linked to circadian translational control of the clock gene *Period1* [47] and localizes to cytoplasmic RNA granules [48]; its strong, stress-dependent induction suggests a coupling between stress signaling and post-transcriptional regulation. Collectively, these observations support the idea that rhythmic variation in specific RBP abundance—and possibly in their post-translational states—can modulate the temporal capacity for SG assembly. Several caveats apply: (i) mRNA rhythms do not necessarily predict protein rhythms (as seen for LARK), (ii) detecting RBP oscillations does not prove their recruitment to SGs at specific times, and (iii) causal roles must be tested. These steps are essential to determine whether the RBPs identified here actively drive SG rhythmicity or merely reflect upstream rhythmic processes.

TTFL-independent circadian oscillations have been described in diverse organisms [49,50,51,52]. Although *Bmal1^−/−^* mice lack canonical rhythms, residual oscillations in protein synthesis persist [53]. Given the rhythmicity of global translation [54,55,56,57], it is plausible that SG formation is modulated by translation initiation complex dynamics. In addition, redox oscillations that persist without TTFLs [5,49,58,59] may also influence SG dynamics. However, these alternative mechanisms were not addressed in our study and remain speculative.

In conclusion, SG dynamics in fibroblasts subjected to oxidative stress induced by sodium arsenite display rhythmic regulation, even in the absence of a functional canonical molecular clock. Given that the formation of these biomolecular condensates is an integral component of the cellular stress response, promoting cell survival, whereas defects in their assembly or clearance have been linked to neurodegeneration and other diseases, the rhythms described here contribute to a deeper understanding of how organisms respond to stress challenges at different times of the day. Future studies are warranted to elucidate the molecular mechanisms underlying these rhythms and to determine their physiological relevance.

## 4. Materials and Methods

### 4.1. Cell Cultures

NIH/3T3 fibroblasts (ATCC) were cultured and synchronized as previously described [17,18,60]. Cells were maintained at 37 °C in a humidified atmosphere of 5% CO_2_ and 95% air, in Dulbecco’s Modified Eagle Medium (DMEM; Gibco, Brisbane, Australia) supplemented with 10% calf serum (CS; Gibco). *wt* and *Bmal1^−/−^* mouse embryonic fibroblast (MEF) lines derived from *wild-type* or *Bmal1^−/−^* mice, respectively, both harboring the PER2::LUC reporter [27]. These cell lines were kindly provided by Dr. Joseph S. Takahashi (UT Southwestern Medical Center, Dallas, TX, USA) and were cultured in DMEM supplemented with 10% fetal bovine serum (FBS; Gibco). Experiments were performed using quiescent cultures (24–48 h after reaching confluence). In fibroblasts, cell division is inhibited by contact; additionally, cultures were maintained in 0.5% serum after confluence, ensuring that cells remained in a quiescent (G_0_) state and that the cell cycle did not progress. For synchronization by serum shock, at T = 0 h (unstimulated cells), culture medium was replaced with pre-warmed DMEM containing 50% horse serum (Gibco) for 2 h, after which it was exchanged for pre-warmed DMEM containing 0.5% CS. Alternatively, cultures were synchronized with 100 nM dexamethasone for 1 h, followed by replacement with pre-warmed DMEM containing 0.5% CS (NIH/3T3) or 0.5% FBS (MEFs). At the indicated time points after stimulation, cells were washed in phosphate-buffered saline (PBS) and processed as follows: fixed for immunocytochemistry (ICC), harvested in TRIzol^®^ reagent (Invitrogen, Carlsbad, CA, USA) for RNA extraction, or lysed in RIPA buffer (50 mM Tris-Cl pH 7.5, 1% IGEPAL CA-630 [Sigma], 0.5% sodium deoxycholate, 0.05% SDS, 1 mM EDTA, 150 mM NaCl) containing a protease inhibitor cocktail (Sigma, Saint Louis, MO, USA) for protein analysis. For stress granule (SG) induction, cultures were treated with 500 µM sodium arsenite for 30 min to induce oxidative stress before harvesting at each time point.

### 4.2. Immunocytochemistry (ICC) and Image Acquisition

Immunodetection of SGs was performed as described by Kedersha and Anderson [20], using a goat anti-eIF3 antibody (Santa Cruz Biotech, Dallas, TX, USA), following our previous protocols [61]. Cells grown on coverslips were treated with sodium arsenite to induce SGs, washed twice in PBS, and fixed in 4% paraformaldehyde-PBS for 15 min at room temperature. They were then permeabilized with −20 °C methanol for 10 min, washed twice in PBS, and blocked in 5% horse serum-PBS for 2 h at room temperature. Primary antibody incubation (1:200 in blocking solution) was carried out for 1 h at room temperature or overnight at 4 °C. After three PBS washes, samples were incubated with donkey anti-goat IgG (1:2000, DyLight™; Jackson ImmunoResearch, West Grove, PA, USA) and Hoechst dye (50 ng/mL) in blocking solution for 30 min at room temperature. Samples were then washed three times in PBS and mounted with FluorSave reagent (Calbiochem, Darmstadt, Germany). Images were acquired on a Zeiss Axioplan fluorescence microscope with a UPlanSApo 60×/1.35 NA oil-immersion objective (Olympus, Tokyo, Japan) and an XM10 monochrome camera (Olympus, Tokyo, Japan), using CellSens Entry software (v4.2. Olympus, Tokyo, Japan). Exposure times were set using the brightest SG signal to ensure that maximum pixel intensity was within the mid-range of the 14-bit CCD sensor, preventing saturation and ensuring a linear response across SG signal intensities.

### 4.3. Stress Granule Quantification

SG quantification was adapted from the method described by Nissan and Parker [62] for PBs, with modifications for SG analysis. All images were processed using algorithms to ensure unbiased treatment across the dataset. Digital images (1376 × 1038 pixels, 8-bit) from each experiment were processed together as a single stack in ImageJ, following these steps:Process > “Smooth”;Process > “Subtract background” (rolling ball radius: 10 pixels);Plugins > Segmentation > “Otsu thresholding”;Analyze > “Analyze particles” (size: 17–1650 pixels^2^; circularity: 0.6–1.0).

For each image, we determined the number of particles, their mean intensity (after background subtraction), and mean area. Nuclei were counted from Hoechst-stained images. With these size restrictions, the mean SG diameter across all time points was 1.28 μm for *wt* cells and 1.11 μm for *Bmal1^−/−^* cells. All measurements are reported as relative values (pixels^2^), as fluorescence microscopy is constrained by diffraction, and image processing involves steps that are, to some extent, arbitrary, such as background subtraction and thresholding during particle delineation.

### 4.4. Western Blotting

Standard Western blotting protocols were followed as previously described [18,60]. Briefly, total protein content in homogenates was determined using the Bradford method. Protein lysates (20 μg) were separated on 12% SDS–polyacrylamide gels and transferred to nitrocellulose membranes (Sigma). The anti-eIF3 antibody was the same as that used for ICC. Primary antibodies were directed against p-eIF2α (Abcam, Waltham, MA, USA), TIA-1 (Santa Cruz Biotech, Dallas, TX, USA), LARK/RBM4 (Protein Tech, Martinsried, Germany), and α-tubulin (Sigma). Detection was performed with IRDye secondary antibodies (LI-COR Biosciences, Lincoln, NE, USA), and membranes were scanned using an Odyssey IR Imager (LI-COR Biosciences, Lincoln, NE, USA).

### 4.5. RNA Isolation, cDNA Synthesis and Real-Time Quantitative PCR

Total RNA was extracted using TRIzol^®^ reagent (Invitrogen) according to the manufacturer’s instructions. One microgram of total RNA was reverse-transcribed using ImProm-II™ Reverse Transcriptase (Promega, Madison, WI, USA) with a combination of oligo(dT) and random hexamer primers, following the manufacturer’s protocol. SYBR Green–based real-time qPCR was performed as previously described [18,60]. Primer sequences (Sigma) are listed in Table 2.

Reactions were run on a Rotor-Gene Q instrument (QIAGEN) in a total volume of 25 μL containing 1 μL of cDNA, 0.5 μM of each primer, and 12.5 μL of Real Mix (Biodynamics, Buenos Aires, Argentine). The cycling protocol consisted of polymerase activation at 95 °C for 30 s, followed by 40 cycles of 95 °C for 30 s, 60 °C for 30 s, and 72 °C for 30 s. Each assay included a duplicate standard curve generated from 1:5 serial dilutions of cDNA from 2 h serum-stimulated cells. All samples were measured in triplicate. Specificity of amplification was verified by melt curve analysis, and both standard curve linearity and PCR efficiency were optimized. Relative transcript abundance was calculated using Rotor-Gene Q software (version 2.1.0.9. QIAGEN, Venlo, The Netherlands) and normalized to the geometric mean of three reference genes: *Tbp*, *B2m*, and *18S rRNA* [68].

### 4.6. Bioluminescence

The bioluminescence of *wt* and *Bmal1^−/−^* cell samples was determined with a Glomax 20/20 luminometer (Promega) by using Luciferase Assay System (Promega) according to manufacturer instructions. Briefly, since *wt* and *Bmal1^−/−^* MEFs already harbor the PER2::LUC circadian reporter (see “Cell culture” Section 4.1), we just harvested the synchronized cells at the indicated times in Passive Lysis Buffer and then proceeded to quantify luminescence in presence of the substrate under luciferase reaction conditions (Luciferase Assay Reagent).

### 4.7. Statistical Analysis

Statistical analyses included analysis of variance (ANOVA) followed by Tukey HSD and Benjamini & Hochberg post hoc corrections. To assess periodicity in time-series data, we employed MetaCycle [24], an R (4.3.3) package that integrates three algorithms: ARSER [69], JTK_CYCLE [70], and Lomb-Scargle [71]. MetaCycle statistically determines whether the data exhibit rhythmicity and estimates period, amplitude, and phase. Additionally, time-series data from Figure 5 were analyzed using Singular Spectrum Analysis (SSA), a model-free statistical method implemented in R that decomposes signals into trend and oscillatory components [25]. Since RSSA by default represents reconstructed components using internal indices rather than experimental time points, we relabeled the x-axis in the plots to display the actual sampling times (8 to 68 h post synchronization, every 4 h). This adjustment, illustrated in Appendix A, was made exclusively for visualization purposes and does not alter the underlying decomposition or reconstructed signals. In cases where we evaluated whether the oscillations were stationary, we applied the augmented Dickey–Fuller (ADF) test using the *urca* library from the *bootUR* package in R.

## Figures and Tables

**Figure 1 ijms-26-09943-f001:**
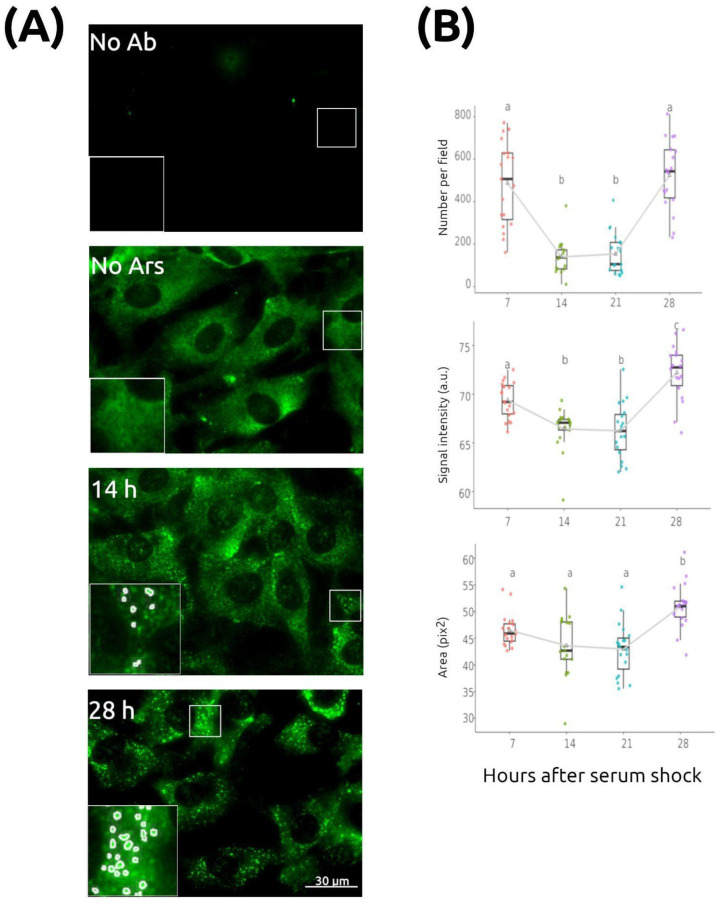
Stress granules change over time independently of the cell cycle. Quiescent NIH/3T3 cultures were synchronized with a serum shock at time 0 and fixed for immunocytochemistry (ICC) at the indicated times. At each time point, SGs were induced with 500 µM sodium arsenite for 30 min before fixation and detected using an anti-eIF3 antibody. (**A**) Representative fluorescence images showing: control with no primary antibody (No Ab), unstressed cells without arsenite treatment (No Ars), and stressed cells fixed 14 h or 28 h after serum shock. Insets show magnified views of selected regions with quantification masks overlaid (see Section 4). Scale bar: 30 µm. (**B**) Quantification of SG parameters (box plots represent the median, interquartile range, and minimum/maximum values within 1.5× IQR, *n* = 17–19 images/group). All three variables—SG number per field, mean eIF3 signal intensity, and SG area—varied significantly over time. Different letters indicate statistically significant differences (*p* < 0.05, Tukey’s post hoc test).

**Figure 2 ijms-26-09943-f002:**
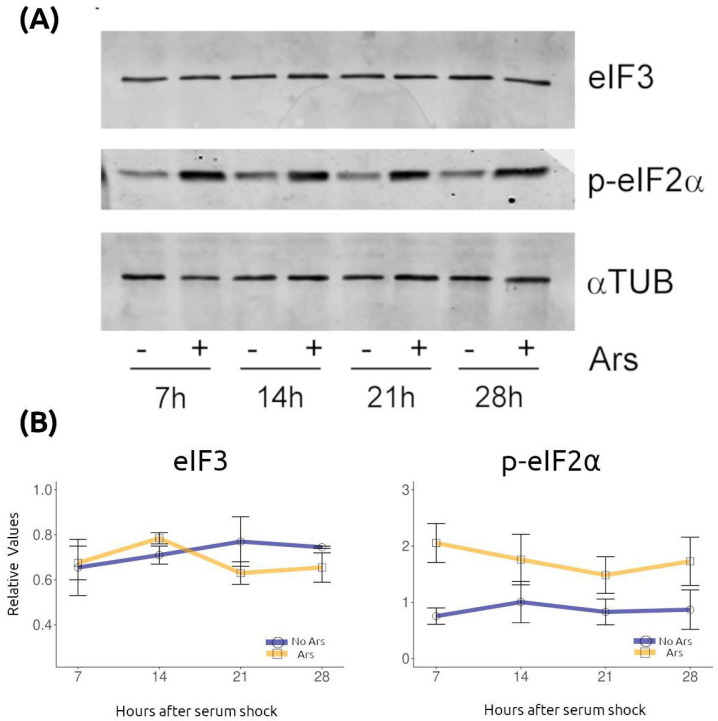
eIF3 protein levels and eIF2α phosphorylation remain unchanged over time. (**A**) Quiescent NIH/3T3 cultures, synchronized and treated as in Figure 1, were harvested at the indicated times and analyzed by Western blot for eIF3, phosphorylated eIF2α (p-eIF2α), and α-tubulin (αTUB, loading control), with or without sodium arsenite treatment (Ars). (**B**) Densitometric quantification of eIF3 and p-eIF2α protein levels, normalized to αTUB, is shown (mean ± SEM from two independent experiments).

**Figure 3 ijms-26-09943-f003:**
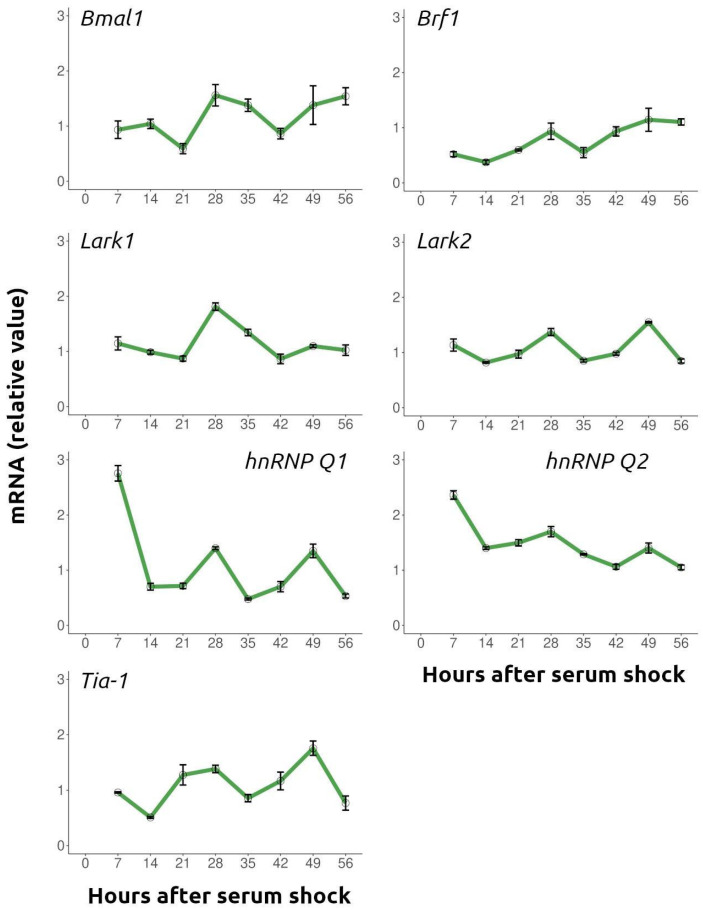
Temporal expression profiles of transcripts encoding RNA-binding proteins associated with cytoplasmic RNA granules. Synchronized NIH/3T3 cultures were harvested at the indicated time points after serum shock (no arsenite treatment). Total RNA was extracted, reverse transcribed, and specific transcript levels were quantified by real-time qPCR. Data represent one representative experiment (mean ± SEM, *n* = 3).

**Figure 4 ijms-26-09943-f004:**
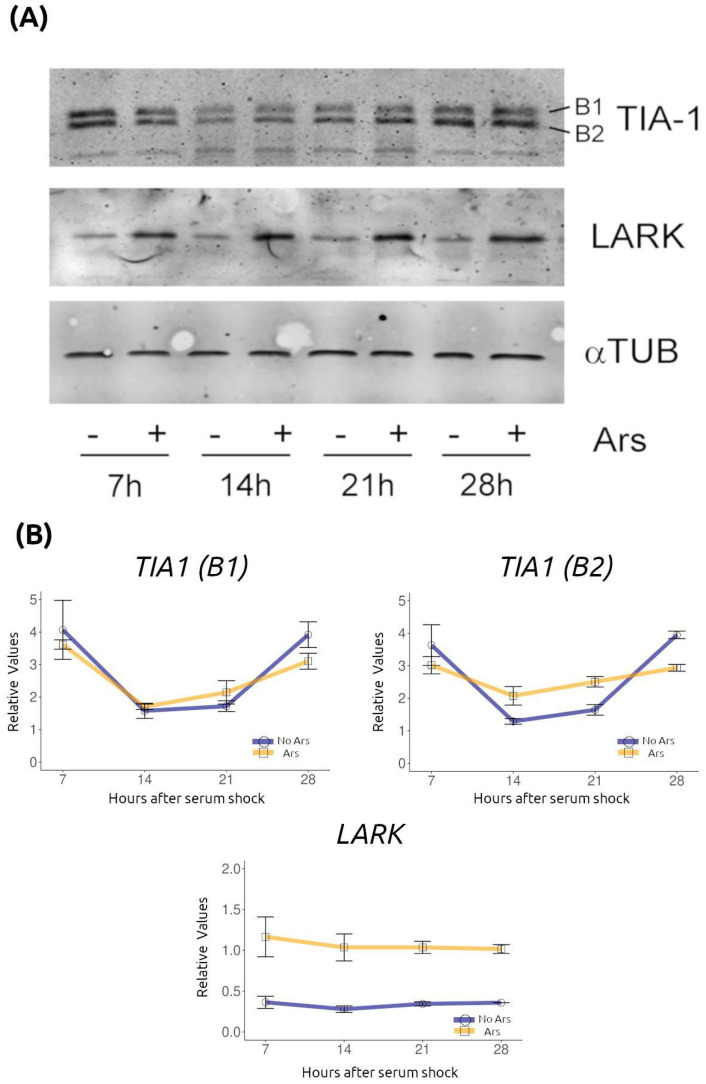
Temporal regulation of TIA-1 protein levels and LARK induction by arsenite. (**A**) Quiescent NIH3T3 cultures, synchronized and treated (+) or not (−) with arsenite as described in Figure 1, were harvested at the indicated time points. Samples were analyzed by Western blot. The anti-TIA-1 antibody detected two bands, B1 and B2, both bands exhibited similar temporal patterns. Representative blots of all three proteins analyzed from the same protein extracts are shown. (**B**) Densitometric quantification of TIA-1 and LARK protein levels, normalized to αTUB, is shown (mean ± SEM from two independent experiments).

**Figure 5 ijms-26-09943-f005:**
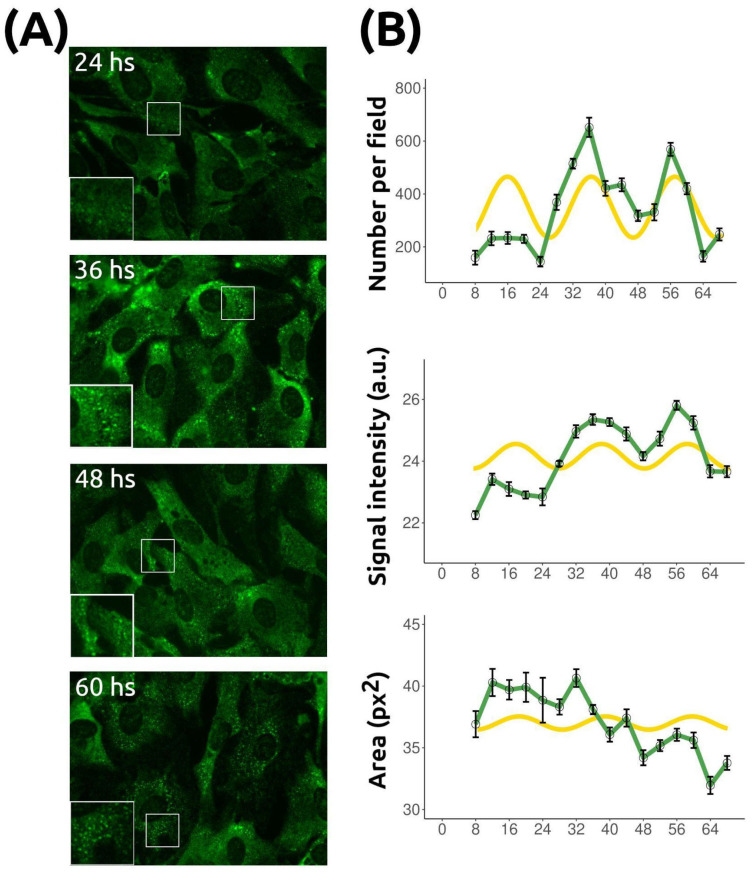
Rhythmic temporal changes in stress granules. Confluent, serum-deprived (quiescent) NIH/3T3 cells were synchronized with dexamethasone. SGs were induced by arsenite at each time point as described in Figure 1. (**A**) Representative images of NIH/3T3 cells at 24, 36, 48, and 60 h after dexamethasone synchronization, followed by SG induction with sodium arsenite, are shown. (**B**) Quantitative analyses revealed time-dependent variations in the number of SGs per field, eIF3 mean fluorescence intensity, and SG area (mean ± SEM). Green curves correspond to experimental measurements; yellow curves represent cosine fits with period, amplitude, and phase estimated by MetaCycle (Table 1). While SG number aligns reasonably well with the fitted function, the fit is poorer for SG signal intensity and especially for SG area, which displays a downward trend over time consistent with the non-stationarity detected by the augmented Dickey–Fuller (ADF) test.

**Figure 6 ijms-26-09943-f006:**
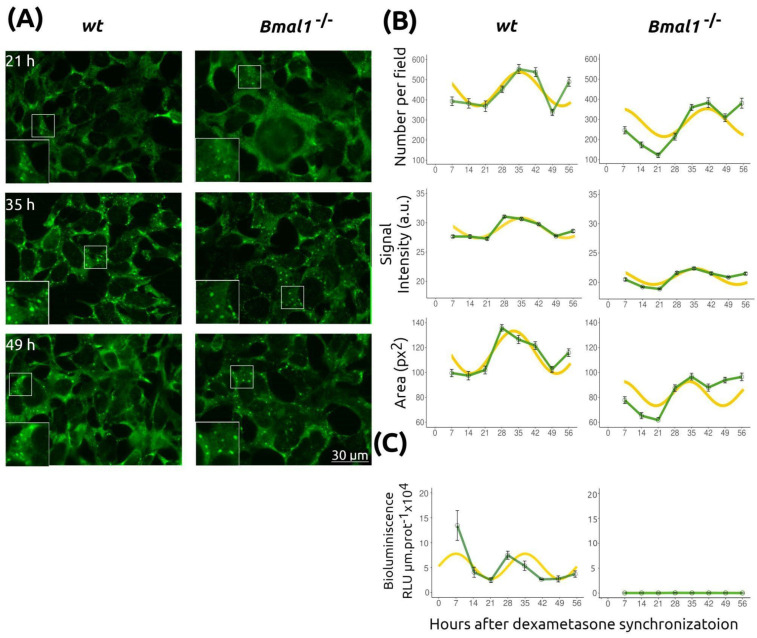
SG oscillations persist in cells with an impaired canonical molecular clock. Confluent, serum-deprived (quiescent) *wt* and *Bmal1^−/−^* MEFs were synchronized with dexamethasone. (**A**) Representative images of *wt* and *Bmal1^−/−^* cells at the indicated time points. Insets show higher-magnification views of selected areas. Scale bar: 30 µm. (**B**) Quantification of SG number per field, signal intensity, and area at different time points, as in Figure 1 and Figure 5 (mean ± SEM). (**C**) Circadian clock activity was assessed by bioluminescence from the PER2::LUC reporter in both *wt* and *Bmal1^−/−^* cells. Green curves correspond to experimental measurements (mean ± SEM); yellow curves represent cosine fits with period, amplitude, and phase estimated by MetaCycle (Table 1).

**Table 1 ijms-26-09943-t001:** Analysis of the temporal changes observed in NIH/3T3, *wt* and *Bmal1^−/−^* MEF cultures.

Cell Type	ANOVA		MetaCycle (meta2d) [24]					*n* ^1^
	*p*	*F*	*p*	BH, Q	Period	Phase	Amp	
1. *Stress granule number per field*								
NIH/3T3	<0.001	38.3	<0.001	<0.001	23.98	21.82	330.8	13
*wt*	<0.001	14.3	3.5 × 10^−14^	3.5 × 10^−14^	34.48	0.67	83.91	55–64
*Bmal1^−/−^*	<0.001	29.8	1.3 × 10^−7^	1.3 × 10^−7^	35.00	6.10	72.77	51–65
2. *Stress granule signal intensity*								
NIH/3T3	<0.001	33.2	<0.001	<0.001	20.08	17.43	28.14	13
*wt*	<0.001	48.4	<0.001	<0.001	35.00	34.84	1.69	55–64
*Bmal1^−/−^*	<0.001	44.5	<0.001	<0.001	33.29	2.05	1.35	51–65
3. *Stress granule area*								
NIH/3T3	<0.001	9.4	0.005	0.005	20.00	8.000	0.566	13
*wt*	<0.001	22.5	<0.001	<0.001	35.00	32.83	17.06	55–64
*Bmal1^−/−^*	<0.001	28.6	1.1 × 10^−12^	1.1 × 10^−12^	27.33	6.60	9.71	51–65
4. *Bioluminescence*								
*wt*	<0.001	8.80	<0.001	<0.001	28.13	4.21	32,207	3
*Bmal1^−/−^*	0.188	1.69	0.516	0.516	-	-	-	3

^1^ Number of images analyzed at each time point; for bioluminescence corresponds to the number of wells that were analyzed at each time point.

**Table 2 ijms-26-09943-t002:** Primer sequences for Real-Time RT-qPCR.

*Abbr.*	*Names*	*Accession Number*	*Forward* *Reverse*	*Ampl. Size*
*18S rRNA*	*18S ribosomal RNA*	X00686	CGCCGCTAGAGGTGAAATTC [60,63] CGAACCTCCGACTTTCGTTCT	101 nt
*B2m*	*β2-Microglobulin*	NM_009735	TTCTGGTGCTTGTCTCACTGA [60] CAGTATGTTCGGCTTCCCATTC	104 nt
*Bmal1*	*Aryl hydrocarbon receptor nuclear translocator-like*	NM_007489	GCAGTGCCACTGACTACCAAGA [64] TCCTGGACATTGCATTGCAT	201 nt
*Brf1*	*zinc finger protein 36, C3H type-like 1 (Zfp36l1)*	NM_007564	TGCGAACGCCCACGAT [65] CTTCGCTCAAGTCAAAAATGG	60 nt
*hnRNP Q1*	*synaptotagmin binding, cytoplasmic RNA interacting protein (Syncrip) 1*	NM_019666	GCCGCGGTGGAAATGTAGGAGG [66] TGATTATTGGTCTGGCGCCGCT	82 nt
*hnRNP Q2*	*synaptotagmin binding, cytoplasmic RNA interacting protein (Syncrip), 2*	NM_019796	CAACAACAAAGAGGCCGCGGG [66] ACCATTACTCCACTGCAAGCTTCTG	116 nt
*Lark1*	*RNA binding motif protein 4 (Rbm4)*	NM_009032	GGTTACGGGCATGACAGTGAG [67] GGCCATGTCGTACAGGGAAT	69 nt
*Lark2*	*RNA binding motif protein 4B (Rbm4b)*	NM_025717	CCAGTAGACCGTACAGGGC [67] GGACTCCCCATAACCCATAGT	99 nt
*Tbp*	*TATA box binding protein*	NM_013684	AGAACAATCCAGACTAGCAGCA [60] GGGAACTTCACATCACAGCTC	120 nt
*Tia-1*	*cytotoxic granule-associated RNA binding protein 1*	NM_011585	GGTGCCTCAAGGATTCCCTGTTGG [66] GCACGAGACTGATGCGGCGA	149 nt

## Data Availability

The images generated and analyzed in this study, as well as other data, are available from the corresponding author upon request.

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
