# Peer review of "Rhythmic Dynamics of Stress Granules in *Wild-Type* and *Bmal1^−/−^* Fibroblasts Lacking a Functional Canonical Circadian Clock"

_ijms, 2025, doi:10.3390/ijms26209943_

Round 1

Reviewer 1 Report

Comments and Suggestions for Authors

The manuscript by Malcolm et al. presents an interesting insight into a particular cellular stress response and its potential circadian regulation. The analysis appears thoroughly conducted and the results well-described and discussed, including the limitations of the study. I would suggest a few improvements:

-since the MetaCycle analysis was applied also to the data from Figure 5, could the authors also show the cosine fits as for Fig 6?

-In the MetaCycle analysis, what does the phase parameter (Table 1) mean? Could they be presented in a way that would allow to compare them between neo (mostly above 30h?) and Bmal1 (<7h). Also the finding of a >30h period in these data could be discussed.

Minor points;

-on Figure 3: please clarify the gene names which are distinct from the text (hnRNP vs hnRNP Q)

-on Figure 4: please provide an explanation/hypothesis for the presence of 2 bands in TIA-1

-recheck the references: the ones of Kornmann and Koronovsky, both indicated as 44 are not in the list

Reviewer 2 Report

Comments and Suggestions for Authors

The finding of a SG rhythmicity which may be independent of the canonical circadian system is very interesting. However, overall, it's too descriptive, more experiments regarding the investigation of possible mechanisms, or the physiological consequence of the rhythmicity, are missing.

Other comments:

1. It is not necessary to show too much details of statistics information in the text, as they are indicated in the methods, figures and figure legends. 
2. Fig. S1: 1) since there are three panels, it will be better to label and list them alphabetically; 2) in the bottom panel, the time length is only 3.5 times 4h = 14 h, why the authors call it a rhythm? It should be at least 24 h or more.
3. Fig. 2, Fig. 4: statistics needs to be shown. 
4. Gene names in Fig. 3 should be italic; the size of some characters is too large and the place of "hours after serum shock" is incorrect.
5. Fig. 6: the labeling of "Bmal1" is wrong, it should be Bmal1-/-.

Round 2

Reviewer 1 Report

Comments and Suggestions for Authors

The authors addressed all my questions and the manuscript was improved appropriately.

Reviewer 2 Report

Comments and Suggestions for Authors

My concerns especially regarding the mechanism issue has not been substantially addressed.